# Trends in Suicide Mortality in 10 Years around the Great East Japan Earthquake: Analysis of Evacuation and Non-Evacuation Areas in Fukushima Prefecture

**DOI:** 10.3390/ijerph18116005

**Published:** 2021-06-03

**Authors:** Yujiro Kuroda, Masatsugu Orui, Arinobu Hori

**Affiliations:** 1Research Department, Fukushima Prefectural Centre for Environmental Creation, Fukushima 963-7700, Japan; 2Sendai City Mental Health and Welfare Center, Sendai 980-0845, Japan; oruima@fmu.ac.jp; 3Department of Psychiatry, Hori Mental Clinic, Fukushima 979-2335, Japan; arinobu.h@gmail.com

**Keywords:** disaster public health, social epidemiology, suicide prevention, social capital

## Abstract

This study analyzed the suicide mortality rate in 12 municipalities in Fukushima Prefecture designated as evacuation areas following the 2011 nuclear disaster. Changes in suicide rates were examined using an exponential smoothing time series model. In the evacuation areas, the suicide rate of men increased immediately after the disaster and then decreased from 47.8 to 23.1 per 100,000 during about 1½ years after the disaster. However, with the lifting of the evacuation order, it again exceeded that of non-evacuation areas and continued to do so for the next 3 years. On the other hand, the suicide rate in women in the evacuation areas increased later than that in men. These results indicate the need for continuous support following the lifting of the evacuation order. In addition, it is necessary to enhance social networks, which continue to confer protection, because of the isolation of the elderly as highlighted in our previous study.

## 1. Introduction

The accident at the Fukushima Daiichi Nuclear Power Plant (FDNPP) following the Great East Japan Earthquake of 11 March 2011 resulted in a triple disaster of earthquake, tsunami, and nuclear disaster, and brought about a major change in the lives of residents in and around Fukushima Prefecture. The level of radiation contamination varies among the 12 municipalities in Fukushima Prefecture that became evacuation zones [1]. In some areas, such as Tamura City and Hirono Town, the evacuation order was lifted within 3 years [2]. However, residents of 10 other municipalities were forced to live as long-term evacuees for more than 5 years. The order has not been lifted in areas designated as difficult to return to across these municipalities as of March 2021. This prolonged evacuation has undermined mental health owing to changes in lifestyle and family role associated with changes in living quarters and family structure, as well as the continued separation of communities [3,4,5,6]. Moreover, the mental health problems faced by the victims change over time [7]. In the immediate aftermath of a disaster, mental health patients continue to receive medical care [8], but people experience traumatic reactions and loss due to the disaster [9,10]. In the medium to long term, problems such as insomnia, depression, and alcohol dependence have been reported [11,12,13], and in the reconstruction period, re-separation of communities and the loneliness associated with moving to a new home have been reported [7,14,15].

As an indicator of the mental health of the disaster victims, Orui et al. monitored suicide rates in the evacuation areas [16,17]. As of the end of 2015, the suicide rates had increased in municipalities where the evacuation order had been lifted relative to municipalities where it had not been lifted. Using National Police Agency data from the disaster-stricken prefectures of Fukushima, Miyagi, and Iwate, Takebayashi et al. reported that suicide mortality remained elevated only in Fukushima [18]. Thus, compared with Miyagi and Iwate, which suffered more damage from the tsunami, Fukushima, which was affected by the nuclear disaster, faced long-term factors contributing to an increase in the suicide rate. Even if the evacuation order is lifted, suicide mortality may increase, so long-term monitoring is necessary.

The lifting of the evacuation order means that former residents can return to their homes, but there are negative aspects also. First, it means the termination of temporary housing, necessitating another move and, in many cases, the loss of the new, established living conditions and the associated communities. Not all evacuees return home, but either way, they can end up feeling isolated and lonely [19]. Another challenge is an increased economic burden. The temporary housing was provided free of charge, along with compensation for psychological burden. After the order is lifted, the temporary housing is closed and compensation is reduced, thereby increasing costs for the evacuees. A third challenge is family conflict. Under a prolonged evacuation order, children’s schools, parents’ work, and other livelihoods are often relocated to the evacuation sites [7,20,21]. In such cases, the decision of whether to return or not is made after the evacuation order is lifted, and disagreements often arise within families. By April 2017, the evacuation order had been lifted in 10 municipalities (include partially difficult-to-return-to zones), but on account of the above three challenges, an increase in the suicide rate seems possible every time an evacuation order is lifted. Therefore, the working hypothesis of this study is that continuous support for residents, measures against suicide, and mental health options will continue to be required. However, no studies have been conducted to monitor the suicide rate in Fukushima Prefecture following the lifting of the evacuation order in several municipalities between March and April 2017.

We examined the trend of the suicide mortality rate in 12 municipalities in Fukushima Prefecture designated as evacuation areas following the nuclear disaster since March 2017. Recent studies have pointed out the similarities between the FDNPP accident and the public response to coronavirus disease 2019 (COVID-19) [22]. Such crises exacerbate the risk of suicide in society as a whole, including problems with employment, livelihoods, and relationships, which can contribute to suicide. Therefore, we also discuss how to use the knowledge obtained in this study for mental health and welfare activities in future disasters and crises.

## 2. Materials and Methods

We used three types of data in a descriptive design. (1) We used monthly suicide data collected in each municipality in Fukushima Prefecture from January 2009 to December 2018. Since the data are centralized in the Ministry of Health, Labor and Welfare (MHLW) of Japan, they were obtained with the official permission of the MHLW under Article 33 of the Statistics Act. Data included age, gender, and address at the time of death. (2) To calculate suicide rates, basic annual population data from the resident registry of each municipality (as of 31 March 2009–2018) were collected from the Ministry of Internal Affairs and Communications’ Statistics Bureau. (3) The initial number of evacuees, the number of people who returned home after the evacuation order was lifted, and the population of returnees aged 65 or older were obtained from data provided by municipalities.

Evacuation zones have been repeatedly reorganized since immediately after the earthquake. We refer to 3 zones designated by the Japanese government in April 2012 on the basis of air dose rates: (a) difficult-to-return-to areas, with a radiation dose rate of ≥50 millisieverts (mSv) per year; (b) residence restriction areas, with a radiation dose rate of ≥20 and <50 mSv per year; and (c) areas where the evacuation order is ready to be lifted, with a radiation dose rate of <20 mSv per year. The residents of these areas were forced to relocate to non-evacuation areas and were not allowed to stay overnight. Currently, difficult-to-return-to areas are still subject to protective measures, such as barricades, on account of high air radiation dose rates. We define 12 designated municipalities (Hirono, Tamura, Naraha, Katsurao, Kawauchi, Minamisoma, Kawamata, Iitate, Namie, Tomioka, Ohkuma, and Futaba) as the “evacuation areas” (with a total population of ~178,000; Figure 1) and other municipalities in Fukushima Prefecture as “outside of the evacuation area” (1.8 million).

Annual suicide rates were calculated for each of the 12 designated municipalities and for the non-evacuation areas by adding the monthly number of suicides in each municipality and dividing the total by the Basic Resident Register population, in order to exclude the effect of seasonal variation of suicide rate (exponential smoothing time series model). Then suicide rates were analyzed by age categories using a 2-year simple moving average because the number of age-categorized suicide cases was relatively small. These analysis methods were adopted with reference to the previous study by Orui et al. [17]. For reference, we calculated the percentage of people who returned to their original municipalities in areas where the evacuation order was lifted and the percentage of people aged 65 years or older (definition of “elderly” in Japan) in those areas.

## 3. Results

### 3.1. Changes in Suicide Rates in Evacuation and Non-Evacuation Areas by the Exponential Smoothing Time Series Model

Table 1 shows basic statistics of the numbers of suicide deaths and suicide rates in evacuation areas, non-evacuation areas, and nationally. Suicide rates were higher in men than in women across all regions. The suicide rate among women was higher in the evacuation areas than outside the evacuation areas or the national average.

Changes in suicide rates were examined by using the exponential smoothing time series model (Figure 2). In the evacuation areas, the suicide rate among men increased from 30.4 in 2010 to 47.8 per 100,000 immediately after the disaster, decreased to that in the non-evacuation areas and the national average within 1 year after the earthquake, and then decreased further to 23.1 per 100,000 within 2½ years. Until 4 years after the earthquake, it remained lower in the evacuation areas than in the non-evacuation areas. However, by 4½ years, with the first lifting of the evacuation order, it exceeded that in the non-evacuation area and continued to do so for the next 3 years.

On the other hand, the suicide rate among women in the evacuation areas slightly decreased in the first 1½ years after the disaster but then increased, surpassing that in the non-evacuation areas and the national average. It peaked at 23.5 per 100,000 at 2½ years and then declined again over the next 1½ years. It increased after the first and third lifting of the evacuation order, but about 6 months later than in men.

### 3.2. Analysis of Suicide Rates by Age Categories in Evacuation and Non-Evacuation Areas

Table 2 shows the suicide rates on a 2-year moving average. The suicide rate among men aged ≥70 years rose sharply immediately after the earthquake and then started to decline after 1 year. However, it rose sharply again from March 2014 to February 2017, and again from March 2017 to December 2018. The rate among men aged <30 years was 11.5 per 100,000 before the earthquake, but increased by 5 to 6 per 100,000 after the earthquake, declining only slowly. On the other hand, the rate among men aged 30 to 49 years did not increase following the disaster, and actually fell. The suicide rate among women aged ≥70 years increased from March 2012 to February 2015. That among women aged <30 years also increased slightly from March 2012 to February 2016. The trends in the non-evacuation areas, however, were different. In particular, rates among both men and women trended downward.

### 3.3. Trends in the Number of Residents in Temporary Housing, the Number of Returnees after the Lifting of the Evacuation Order

Figure 3A shows the monthly trends of living situation divided roughly into temporary housing, private rental apartments, and public housing. The largest number of residents was 98,207, in March 2012, of whom ⅓ were in temporary housing. Since then, the number of people living in temporary housing and rental apartments continued to decline, in line with the construction of houses in new towns and the return of residents after the lifting of the evacuation order. A notable decrease followed the lifting of the order in Iitate, Kawamata, Namie, and Tomioka from March to April 2017, at −53.57% year-on-year (Figure 3B). In Hirono and Tamura, where the evacuation order was lifted early, the rate of return was high (>84%; Figure 1). In areas where the order was lifted later, it ranged widely between 8.6% and 51.8%. In addition, population decline and aging are major issues in areas designated as evacuation zones. The pre-earthquake aging rate (percentage of people aged 65 and over) was 24.8% in 2011 and increased to 26.9% in 2014, which is higher than the national rate of 25.2%. For the evacuation zone, there was a substantial increase (ranging from 31.4% to 61.0%), as shown in Figure 1.

## 4. Discussion

We monitored suicide rates in the areas evacuated following the accident at the FDNPP in the medium to long term. The main result is that the suicide rate among men, which increased rapidly immediately after the earthquake and then tended to be restrained, increased again every time the evacuation order was lifted after 2015. Orui et al. revealed an immediate increase in suicide rates only among men in evacuation areas after the nuclear accident [17]. After a temporary decline, overall numbers again increased slightly, but the rate varied by gender, age, and region. A study of the Chernobyl nuclear accident reported a long-term increase in suicide rates among cleaners [23]. In the case of the Great Hanshin-Awaji Earthquake, a large-scale disaster in Japan in 1995, the rate decreased immediately after the earthquake but then increased between 1½ and 2 years later [24,25,26]. Here, we also observed an immediate increase, but the lifting of the evacuation order caused another rise in the suicide rate in men. Women tended to have a later increase in suicide mortality than men, maybe because women have more roles than men (e.g., working, raising children, managing household budget, helping at school) and therefore have more learned resilience, but more detailed qualitative research on gender is needed. At least in Miyagi Prefecture, which was devastated by the tsunami, the suicide rate of women rose later than that of men, as in previous studies [27]. The number of residents in temporary housing did not start to decrease for several months after the lifting of the evacuation order (Figure 3B). During this period, evacuees were forced to make difficult choices of whether to return or not, which increased their psychological burden. The lifting of the evacuation order (over wider areas than before) from March to April 2017 is thought to have triggered a decline in the number of evacuees and a decline in social networks [28]. These results suggest that when life changes once again after a disaster, such as with the lifting of evacuation orders, it is necessary to screen people at risk of suicide at an early stage and link them to support mechanisms, and long-term follow-up is necessary for women. In particular, after the lifting of the order, people become dispersed, making it difficult for health professionals to keep track of residents’ health. “Gatekeepers” who are familiar with the region and its people will, therefore, be useful for monitoring and require effective training.

The secondary result is that the suicide rate among men aged ≥70 years rose sharply immediately after the earthquake and then started to decline a year later. However, it rose sharply again from March 2014 to February 2017 and again from March 2017 to December 2018. We attribute the former increase to the lifting of the evacuation order, and the latter to an increase in the economic burden due to the termination of psychological compensation at the end of FY 2018. Previous studies have shown that depression, social problems, chronic interpersonal difficulties, and family conflicts are risk factors associated with suicidal ideation and attempts in older adults [29,30,31,32]. The drastic changes in life associated with forced evacuation can cause psychological disorders and social problems and increase the suicide rate among the elderly. Our previous study showed that continuous psychological support was necessary for villagers in Iitate Village whose evacuation order was lifted, and that impaired social support resulted from the loss of continuous support [7]. Even if the affected elderly want to return to their hometowns, they may experience serious psychological distress because their houses were damaged during the long evacuation period and the social support they could receive from close relatives and the community is not available due to the fact that young people do not return to their hometowns and there is a lack of welfare services such as home care in their hometowns [19]. There is also a case report of an elderly female evacuee who developed dementia at her evacuation site and showed remarkable abnormal behavior after the evacuation order in her hometown was lifted and her temporary housing was closed. It is reported that the main reason for this is the loss of psychological ties that had been established while they were in temporary housing [33].The percentage of returnees in areas where the evacuation order was lifted early was >84%, but dropped greatly in areas where the order was lifted later (to 8.6–57.7%; Figure 1), indicating that local communities have become fragmented. Therefore, we highlight the following three psychosocial and economic issues that follow the lifting of the evacuation order. It is possible that: (1) the end of the free provision of temporary housing increased the vulnerability of economically disadvantaged people, resulting in an increase in the suicide rate in the community; (2) loss of the communities created in temporary housing during evacuation aggravated mental health; and (3) the exemption of medical expenses for the victims was terminated, making it difficult for them to continue medical care, resulting in mental and physical deterioration. Another important aspect is compensation for psychological burdens. Economic compensation is necessary to rebuild lives, and economic support protects against suicidal intentions. However, compensation can decrease the motivation to work [34]. It is also necessary for government to implement measures to rebuild people’s lives without relying on compensation, in anticipation of the end of compensation following the lifting of the evacuation order.

The suicide rate of men aged 30 to 69 years in the evacuation zone decreased rapidly but that of women did not, even after the evacuation order was lifted. Male suicide rates have been associated with economic conditions such as increased unemployment and bankruptcies [35,36]. Long-term psychological compensation for evacuees is thought to have supported the economy, but another characteristic is that this group of men has been able to resume their lives in their new homes without returning, even after the evacuation order is lifted [37]. They often rebuild their lives in urban areas where there are not only job options but also child education and health care [38]. These factors may have acted protectively in this group of men. Many people who returned to their homes after the order was lifted are elderly, and there is a possibility that they will create a “marginal village”, which is a village with over half the residents over the age of 65, in danger of disappearing. On the other hand, a preliminary survey showed that most of the young people who decided not to return would like to maintain a relationship with their hometowns [19]. To maintain the social network of the elderly in their hometown, it is necessary to support their relatives to have residences in two places.

## 5. Conclusions

In areas where the evacuation order was issued, lifting the order could disrupt lives once again and once more increase the suicide rate. This indicates the need for continuous support. In addition, it is necessary to enhance social networks, which are protective factors, even after the evacuation order is lifted because of the problem of the isolation of the elderly. Specifically, it is necessary to promote not only the activities of the affected residents, but also the networking of the entire region, such as with the original local communities and organizations, in addition to linkages among the affected residents. This builds trust between residents and social capital at various levels.

There are several limitations in this study. First, some evacuees may have changed addresses after the disaster, and the data obtained may not be able to track them and may thus underestimate the suicide rate in the affected areas. In addition, although it is important to know where a person committed suicide after the lifting of the evacuation order (original community or place of refuge), such information could not be obtained. Second, because of the descriptive design of published data, relevant factors such as pre-suicide economic and psychological variables cannot be adjusted. In addition, since it is not clear why the suicide rate of women rises with a delay, it is necessary to supplement this with case studies in the future. Thirdly, since this study adopted a two-year moving average model, the results of the analysis by age group must be interpreted with caution, especially since the data from March 2010 to February 2012 include data from the period when the suicide mortality rate was high after the earthquake. Continuous monitoring of the suicide rate is necessary, and promoting the sharing of results with local government will improve further measurements.

## Figures and Tables

**Figure 1 ijerph-18-06005-f001:**
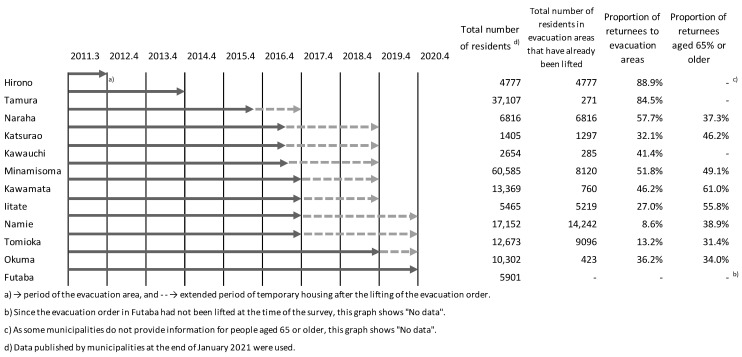
Periods of evacuation order and information on population dynamics.

**Figure 2 ijerph-18-06005-f002:**
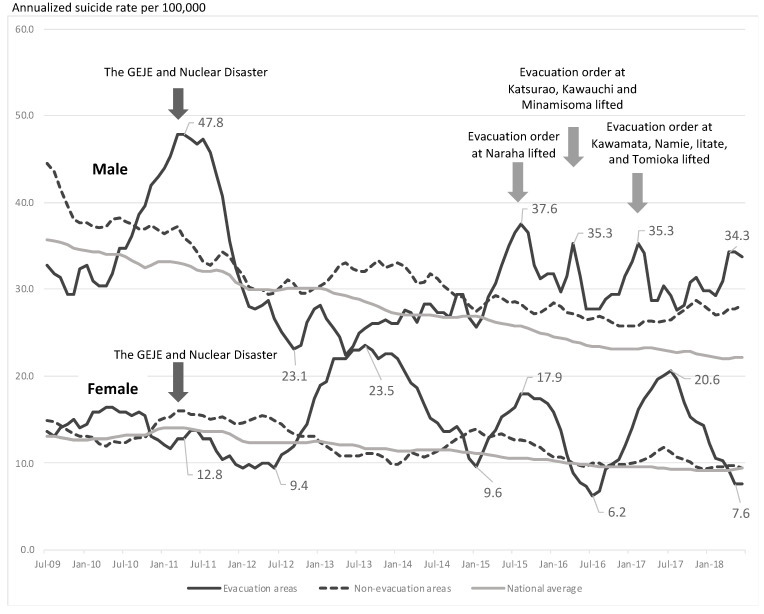
Changes in suicide rates in evacuation and non-evacuation areas by the exponential smoothing time series model.

**Figure 3 ijerph-18-06005-f003:**
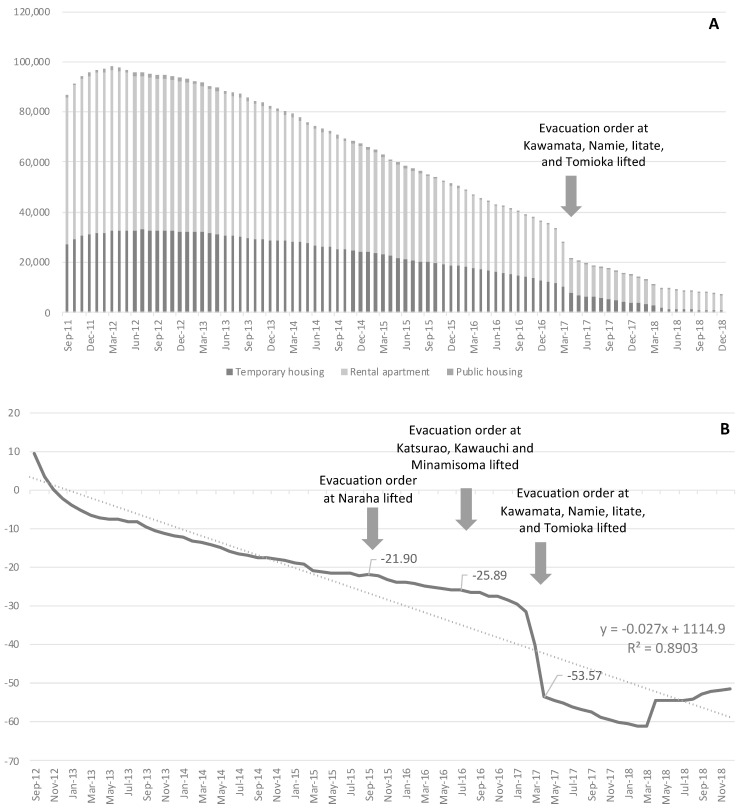
(**A**). Monthly trends of temporary housing residents; (**B**). Monthly trends of temporary housing residents (YoY change, %).

**Table 1 ijerph-18-06005-t001:** Numbers and rates of suicide by month in evacuation areas, areas outside those, and nationwide (from March 2009 to December 2018).

Region	Male			Female		
	Median	Max	Min	Median	Max	Min
Evacuation areas						
Numbers of suicide deaths	2	8	0	1	5	0
Suicide rate	29.8	47.8	22.4	14.1	23.5	6.2
Non-evacuation areas						
Numbers of suicide deaths	23	46	11	9	23	3
Suicide rate	30.3	44.5	25.7	12.2	15.9	9.3
National average						
Numbers of suicide deaths	1442	2180	979	628	1031	383
Suicide rate	27.8	42.1	18.8	11.5	7.0	18.9

**Table 2 ijerph-18-06005-t002:** Suicide mortality rate by age group in both evacuation and non-evacuation areas.

Variables	Male				Female			
	Suicide Mortality Rate (per 100,000)							
Area/Period	<30	30–49	50–69	≥70	<30	30–49	50–69	≥70
Evacuation areas								
2009.3–2011.2 (pre-disaster)	11.5	41.2	42.0	33.1	1.8	8.8	22.3	24.9
2010.3–2012.2 (peri-disaster)	16.8	42.0	42.1	54.3	1.8	9.1	15.5	29.0
2011.3–2013.2 (post-disaster)	17.8	32.7	40.9	54.5	0.0	4.8	19.2	21.6
2012.3–2014.2	14.7	22.1	36.1	39.2	3.9	4.8	22.7	35.2
2013.3–2015.2	18.9	20.2	32.9	41.9	10.1	4.9	15.8	40.9
2014.3–2016.2	17.5	33.9	29.8	56.3	8.3	12.5	16.0	25.2
2015.3–2017.2	18.1	31.9	33.0	56.5	4.3	10.2	19.6	17.5
2016.3–2018.2	16.6	20.7	36.3	36.0	6.6	5.2	21.4	15.8
2017.3–2018.12	12.9	21.1	31.9	55.9	4.6	5.3	12.7	19.8
Non-evacuation area								
2009.3–2011.2 (pre-disaster)	16.3	48.7	56.8	43.1	5.7	13.2	18.1	20.0
2010.3–2012.2 (peri-disaster)	16.2	41.0	48.1	39.1	5.5	13.6	17.7	20.3
2011.3–2013.2 (post-disaster)	16.5	34.5	40.9	41.5	6.1	16.0	16.3	19.7
2012.3–2014.2	17.4	32.1	39.2	43.9	6.0	12.0	12.1	20.2
2013.3–2015.2	16.9	32.9	41.0	36.9	4.8	8.8	11.6	21.5
2014.3–2016.2	14.5	32.9	36.6	35.3	3.2	11.4	14.8	20.4
2015.3–2017.2	13.7	33.6	30.9	34.8	3.2	11.1	12.2	19.4
2016.3–2018.2	11.1	35.0	30.6	36.7	4.8	8.5	10.0	17.5
2017.3–2018.12	11.1	32.0	28.4	33.6	5.4	7.3	9.6	16.2

## Data Availability

The data that support the findings of this study are available from the corresponding author, upon reasonable request.

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
