# Peer review of "Trends in Suicide Mortality in 10 Years around the Great East Japan Earthquake: Analysis of Evacuation and Non-Evacuation Areas in Fukushima Prefecture"

_ijerph, 2021, doi:10.3390/ijerph18116005_

Round 1

Reviewer 1 Report

The manuscript reports on a examination of the suicide rates among survivors of the Tsunami and nuclear disaster at Fukushima Prefecture in 2011. It was found that both men and women who were evacuated from the area had increased rates of suicide, but the elevation in rates for women occurred later than that of the men. Examination of social factors, available support and understanding of the gender roles and responsibilities may explain this difference, but more research is needed to fully comprehend the phenomenon. The article is well written and the results presented clearly in the charts and summary. The subject matter is interesting and could be generalized to other natural or man made disasters in which persons find it necessary to move from familiar areas in uncertain conditions. As many questions were raised as were answered by the report, indicating that further and confirmatory studies should be undertaken. 

Author Response

We thank you for your careful peer review of our paper. As the reviewer pointed out, social factors, available support, gender roles, etc. need to be considered in order to further understand and address this important phenomenon. These variables, which were not clarified in this paper, will be the topic of my next paper. Once again, thank you very much for your thoughtful suggestions and insights

Reviewer 2 Report

This is an interesting and important report about the effects of a natural disaster coupled with a man-made disaster at a nuclear power plant that directly impacted 178,000 people and indirectly affected 1.8 million people.  Natural disasters provide an unwelcomed opportunity to better understand how to protect public health in the event of future disasters.

The background is well-written and explains the motivation for the study.  Figure 1 is helpful in understanding the differing timelines for the 12 affected municipalities.

The time series data was collected from March, 2009, and tsunami occurred two years later in March of 2011. Why was it decided to start in March 2009 and not have more years to analyze given suicide is a rare event with small numbers? 

Related to the above comment, with about 10 years of data and using a 2-year moving average model, does that create only about five time points in the analysis? It seems like a Poisson or Negative Binomial model might be a better way to analyze the data with the inclusion of covariates and predictors.  How was the moving average decided upon?  Were other models considered through fitting processes using differencing adjustments, ACF, PACF and residual analysis?  More detail on the methods and reasons for choosing these methods is needed.

How would you interpret the moving average model in the context of suicide since it simply means that the error in the previous observation helps explain suicide in the next two years?  Was it a moving average of order one or two?

Was an interrupted time series analysis considered using an earlier starting point?

There are two natural groups here – the evaluation area group and the non-evacuation area group.  It seems like more could be done to understand the differences in these groups as they both experienced the event but were differentially impacted by the event. The major difference is the disruption to their lives over a period of time.  This could be explored to assess the effect sizes in the evacuated group.

On page 5, lines 164-165, can you clarify this sentence?

Were there differences in socioeconomic status among residents in the evacuated and not evacuated municipalities?

The figures are well-done and easy to read.  The breakdown and analysis by age group is also interesting and informative.

The references are not complete and need editing.

Author Response

This is an interesting and important report about the effects of a natural disaster coupled with a man-made disaster at a nuclear power plant that directly impacted 178,000 people and indirectly affected 1.8 million people.  Natural disasters provide an unwelcomed opportunity to better understand how to protect public health in the event of future disasters.

The background is well-written and explains the motivation for the study.  Figure 1 is helpful in understanding the differing timelines for the 12 affected municipalities.

The time series data was collected from March, 2009, and tsunami occurred two years later in March of 2011. Why was it decided to start in March 2009 and not have more years to analyze given suicide is a rare event with small numbers? 

> We thank you for your careful peer review of our paper. As pointed out by the reviewers, the impact of this natural disaster and the man-made disaster at the nuclear power plant will have direct and indirect effects on the people in the affected areas and will provide important insights into the public health domain to prepare for future disasters. Since we assume that some international readers may not be familiar with this disaster and the situation in the affected areas, we have added an explanation as shown in Figure 1. Regarding the reviewer's second point, in order to compare with the trends before the GEJE, the observation period was set from March 2009 as a baseline.

Related to the above comment, with about 10 years of data and using a 2-year moving average model, does that create only about five time points in the analysis? It seems like a Poisson or Negative Binomial model might be a better way to analyze the data with the inclusion of covariates and predictors.  How was the moving average decided upon?  Were other models considered through fitting processes using differencing adjustments, ACF, PACF and residual analysis?  More detail on the methods and reasons for choosing these methods is needed.

> As the reviewer pointed out, we agree that other models would have provided a deeper understanding. There are two reasons why this study adopted the two-year moving average model. First, we adopted the same analytical procedures referring to the previous study by Orui et al. (2018). The first point has been added in the paper. The model pointed out by the reviewers will be used in the detailed analysis in the next paper.

- Orui M, Suzuki Y, Maeda M, Yasumura S. Suicide Rates in Evacuation Areas after the Fukushima Daiichi Nuclear Disaster: A 5-Year Follow-Up Study in Fukushima Prefecture. Crisis. 2018;39(5).

How would you interpret the moving average model in the context of suicide since it simply means that the error in the previous observation helps explain suicide in the next two years?  Was it a moving average of order one or two?

> Thanks for the question to clarify the interpretation of the results. We found a steep decrease in suicide rates among middle-aged male adults, especially those aged 30-69 years in evacuation areas, but not in females. Suicide rates among men are reportedly associated with economic circumstances, such as increased unemployment and bankruptcy (Aihara et al., 2003, Yamasaki et al., 2008). One study regarding the 2008 global economic recession and suicide reported that men aged 45-64 years were most affected in European and North and South American countries (Chang et al., 2013). Therefore, the decline in suicide rates in this age category may have contributed to the declining male suicide rates.

- Aihara, H., & Iki, M. (2003). An Ecological Study of the Relations between the Recent High Suicide Rates and Economic and Demographic Factors in Japan. J. Epidemiol., 13(1), 56-61.
- Chang, S. S., Stuckler, D., Yip, P. & Gunnell, D. (2013). Impact of 2008 global economic crisis on suicide: time trend study in 54 countries. BMJ, 347, 1-15.
- Yamasaki, A., Araki, S., Sakai, R., Yokoyama, K. A., Scott Voorhees K. A. (2008). Suicide Mortality of Young, Middle-aged and Elderly Males and Females in Japan fourthe Years 1953–96: Time Series Analysis fourthe Effects of Unemployment, Female Labour Force, Young and Aged Population, Primary Industry and Population Density. Industrial Health 46: 541-549.

Was an interrupted time series analysis considered using an earlier starting point?

> As pointed out by the reviewers, the high suicide mortality rate at baseline (2009-2011) is due to the inclusion of baseline figures in the analysis of the two-year moving average, which is not misleading for those familiar with the results of the two-year moving average analysis. For readers who are not familiar with the results of the two-year moving average analysis, I have added an explanation to the discussion and limitations of the paper.

L275-278

“Thirdly, since this study adopted a two-year moving average model, the results of the analysis by age group must be interpreted with caution, especially since the data from March 2010 to February 2012 includes data from the period when the suicide mortality rate was high after the earthquake.”

There are two natural groups here – the evaluation area group and the non-evacuation area group.  It seems like more could be done to understand the differences in these groups as they both experienced the event but were differentially impacted by the event. The major difference is the disruption to their lives over a period of time.  This could be explored to assess the effect sizes in the evacuated group.

> As the reviewer pointed out, the two groups indicated experienced the same event, but had different impacts from the event. Due to the limitations of the data provided in this study, it is not possible to examine the impact on the level of living between these groups, but the authors have conducted several case studies in the affected areas, which we have cited to strengthen our argument.

L218-227

“Even if the affected elderly want to return to their hometowns, they may experience serious psychological distress because their houses were damaged during the long evacuation period and the social support they could receive from close relatives and the community is not available due to the fact that young people do not return to their hometowns and there is a lack of welfare services such as home care in their hometowns(19). There is also a case report of an elderly female evacuee who developed dementia at her evacuation site and showed remarkable abnormal behavior after the evacuation order in her hometown was lifted and her temporary housing was closed. It is reported that the main reason for this is the loss of psychological ties that had been established while they were in temporary housing(33)”

- Kuroda Y, Koyama Y, Sato N. Farming as a purpose of life as well as a business: Rethinking the reconstruction of food and agriculture in Fukushima after the nuclear accident. In: IAEA Consultancy meeting. 2019.

- Hori A, Ozaki A, Murakami M, Tsubokura M. Development of Behavior Abnormalities in a Patient Prevented from Returning Home after Evacuation following the Fukushima Nuclear Disaster: Case Report. Disaster Med Public Health Prep. 2020;(May):1–4.

On page 5, lines 164-165, can you clarify this sentence?

> Following the reviewer's suggestion, I added an explanation to the sentence about aging in order to clarify the argument.

L164-168

“In addition, population decline and aging are major issues in areas designated as evacuation zones. The pre-earthquake aging rate (percentage of people aged 65 and over) was 24.8% in 2011 and increased to 26.9% in 2014, which is higher than the national rate of 25.2%.For the evacuation zone, there was a substantial increase (ranging from 31.4% to 61.0%), as shown in Figure 1.”

Were there differences in socioeconomic status among residents in the evacuated and not evacuated municipalities?

> There was a disparity among the 12 municipalities that were evacuated. That is, the areas where nuclear power plants were located (Okuma and Futaba towns) had nuclear power plant-related jobs and revenues coming into the municipalities, while the rest of the evacuated areas were often located in rural areas (e.g. Iitate and Kawamata town Yamakiya) and did not depend on the economy for their livelihood. Although there are disparities among the evacuated areas, we believe it is difficult to compare differences in socio-economic status in general because of differences with non-evacuated municipalities. In addition, after the disaster, compensation was made for buildings and land as well as psychological compensation, which is discussed in the text.

The figures are well-done and easy to read.  The breakdown and analysis by age group is also interesting and informative.

> In this paper, we have already used four figures and two tables to make it as visually clear as possible. As pointed out by the reviewers, we considered creating a figure stratified by age group, but we have already presented the results in a table and explained them in the discussion. Since we were concerned that adding a new figure would result in an excessive amount of information, we did not add it this time. In future analyses along the age groups, we would like to introduce the figures suggested by the reviewers.

The references are not complete and need editing.

> The cited references have been revised according to the reviewers' suggestions.